# Identification of an Orally Bioavailable, Brain-Penetrant Compound with Selectivity for the Cannabinoid Type 2 Receptor

**DOI:** 10.3390/molecules27020509

**Published:** 2022-01-14

**Authors:** Meirambek Ospanov, Suresh P. Sulochana, Jason J. Paris, John M. Rimoldi, Nicole Ashpole, Larry Walker, Samir A. Ross, Abbas G. Shilabin, Mohamed A. Ibrahim

**Affiliations:** 1National Center for Natural Products Research, School of Pharmacy, The University of Mississippi, University, MS 38677, USA; mospanov@olemiss.edu (M.O.); lwalker@olemiss.edu (L.W.); sross@olemiss.edu (S.A.R.); 2Department of Chemistry and Technology of Organic Substances, Natural Compounds and Polymers, Al-Farabi Kazakh National University, al-Farabi Ave. 71, Almaty 050040, Kazakhstan; 3Department of BioMolecular Sciences, School of Pharmacy, University of Mississippi, University, MS 38677, USA; spsuloch@olemiss.edu (S.P.S.); parisj@olemiss.edu (J.J.P.); jrimoldi@olemiss.edu (J.M.R.); nmashpol@olemiss.edu (N.A.); 4Department of Chemistry, East Tennessee State University, Johnson City, TN 37614, USA

**Keywords:** cannabinoid receptors CB1/CB2, pyrrolobenzodiazepines, neuroinflammation, neurodegenerative diseases, central nervous system (CNS), pharmacokinetics (PK), radioligand binding assay

## Abstract

Modulation of the endocannabinoid system (ECS) is of great interest for its therapeutic relevance in several pathophysiological processes. The CB2 subtype is largely localized to immune effectors, including microglia within the central nervous system, where it promotes anti-inflammation. Recently, a rational drug design toward precise modulation of the CB2 active site revealed the novelty of Pyrrolo[2,1-c][1,4]benzodiazepines tricyclic chemotype with a high conformational similarity in comparison to the existing leads. These compounds are structurally unique, confirming their chemotype novelty. In our continuing search for new chemotypes as selective CB2 regulatory molecules, following SAR approaches, a total of 17 selected (S,E)-11-[2-(arylmethylene)hydrazono]-PBD analogs were synthesized and tested for their ability to bind to the CB1 and CB2 receptor orthosteric sites. A competitive [^3^H]CP-55,940 binding screen revealed five compounds that exhibited >60% displacement at 10 μM concentration. Further concentration-response analysis revealed two compounds, **4k** and **4q**, as potent and selective CB2 ligands with sub-micromolar activities (*K*_i_ = 146 nM and 137 nM, respectively). In order to support the potential efficacy and safety of the analogs, the oral and intravenous pharmacokinetic properties of compound **4k** were sought. Compound **4k** was orally bioavailable, reaching maximum brain concentrations of 602 ± 162 ng/g (p.o.) with an elimination half-life of 22.9 ± 3.73 h. Whether administered via the oral or intravenous route, the elimination half-lives ranged between 9.3 and 16.7 h in the liver and kidneys. These compounds represent novel chemotypes, which can be further optimized for improved affinity and selectivity toward the CB2 receptor.

## 1. Introduction

Neurodegenerative disorders are a widespread cause of morbidity and mortality worldwide, characterized by the slow, progressive damage/loss of neurons in the central nervous system (CNS), which is associated with deficits in function (e.g., movement, memory, cognition) that are related to the affected CNS region(s) [1]. The progression of many neurodegenerative diseases is thought to be driven by the template-directed misfolding, seeded aggregation, and cell–cell transmission of characteristic disease-related proteins, leading to the consecutive spreading of pathological protein aggregates [2]. Modulating the activity of endocannabinoids (ECB) has held therapeutic promise for treating a wide range of diseases associated with inflammation and oxidative stress. The cannabinoid receptors subtype 2 (CB2), which were identified molecularly in 1993, have been the subject of considerable attention, primarily due to their promising anti-inflammatory potential without the adverse psychotropic effects more commonly associated with CB1 receptor-based therapies [3]. Many studies have further corroborated the in vivo link between chronic neuroinflammation and CB2 upregulation in animal models of pain [4] and inflammation [5]. These studies raise the possible logical connection between CB2 receptors and immunological function [6]. Activation of CB2 receptors by ligands favors a range of receptor conformations that can affect different signaling pathways [7].

Administration of the CB2 antagonist was able to decrease the neuroprotective effects of beta-caryophyllene (BCP) and strengthen the need for CB2 activation. The distribution of the CB2 subtype in neural cell types (activated astrocytes, reactive microglia, oligodendrocytes, perivascular microglia, and neural progenitor cells), many of which are integral to blood–brain barrier function, supports its potential role in CNS integrity [7]. CB2 agonists have shown promising neuroprotection using in vitro (BV-2 microglial cell lines and primary microglia cultures) and in vivo models (murine model) of neurodegeneration. The potential of CB2 agonists as neuroprotective agents has been particularly sought given the lack of psychotropic adverse effects generally seen with CB1 agonists [8]. One of the limitations of the small molecule modulators of CB2 has been issues with selectivity and differential activity depending on cell type/species. Several ‘selective’ CB2 agonists have been shown to regulate orphan cannabinoid receptors and/or opioid receptors or other off-target proteins. In addition, recent evidence suggests that inverse agonists of CB2 may also be neuroprotective. A synthetic CB2 inverse agonist protects neurons from excitotoxic insult in vitro and in vivo [9]. SMM-189 prevented the rat primary microglia-mediated inflammation and was neuroprotective against N-methyl-D-aspartic acid (NMDA)-induced neuronal excitotoxicity in brain-mimicking neuron-glia primary cultures obtained from rat hippocampal tissues [9]. Concurrent development of novel, selective CB2 inverse agonists, partial, and full agonists would allow for a closer comparison of the effects of these two mechanistically distinct pharmacological effects so the role of CB2 in neuroinfectious diseases can be elucidated.

Novel synthetic and natural compounds with neuroprotective potential have emerged as new pharmaceuticals in the management of many neurological conditions, presenting a heterogeneous chemical structure and varied pharmacological activity, which may be useful in the treatment of high-complexity diseases. The search for new drug leads that target endocannabinoid signaling is challenging as endocannabinoids are implicated in various physiological and pathological processes.

Pyrrolo[2,1-c][1,4]benzodiazepines (PBDs) are a class of natural products known to possess anti-tumor and antibiotic activities [10,11,12]. Molecular modeling studies have shown that the PBD tricyclic structure is well-suited to dock within the catalytic cavity of CB receptors, allowing them to interact with the endocannabinoid signaling system. As a part of our continuing efforts to identify novel neuroprotective agents through a structure-based rational design and multi-step synthetic approach [13], the present project focused on the synthesis of seventeen analogs of (S,E)-11-[2-(arylmethylene)hydrazono]-PBD ligands (**4a**–**4q**) as potential CB2 selective regulators that may serve as novel and effective neuroprotectants (Figure 1). Discovery of novel regulatory compounds using structure–activity relationship (SAR) studies will advance the field of neuroprotection and neurorestoration by providing information necessary to determine if this class of compound is a viable chemotype for the design of potentially new therapies to treat neurodegenerative disorders in humans.

Drug discovery programs routinely perform pharmacokinetic (PK) studies in mice to prioritize lead compounds based on anticipated exposure–efficacy and exposure–toxicity relationships. Promising lead compounds with selective binding affinities were examined for their pharmacokinetics properties in plasma or tissue (brain, liver, or kidney) following oral or intravenous administration.

## 2. Material and Methods

### General Experimental Procedures

The ^1^H and ^13^C NMR spectra were recorded in DMSO-*d_6_* and CDCl_3_ on Bruker 400 and 500 MHz spectrometer operating at 400, 500 MHz for ^1^H and 100, 150 MHz for ^13^C NMR. Chemical shift (*δ*) values are presented in ppm and in reference to the residual solvent signals of DMSO-*d_6_* and CDCl_3_ at *δ*_H_/*δ*_C_ 2.50/39.5 and 7.25/70.2, respectively. The coupling constants value (J) reported in Hz.

LC analysis was conducted using an Agilent 1100 HPLC system, RP-C18 column (150 × 4.6 mm; particle size 5 μm; Luna) with column oven temperature set at 25 °C and a gradient system of eluent water (A) and acetonitrile (B). The gradient condition was as follows: 0–2 min (5% B), 2–5 min (5% B→50% B), 5–10 min (50% B→100% B), 10–15 min (100% B). The flow rate of the solvent was 1.0 mL/min, and the injection volume was 25 μL. All analysis was carried out at wavelength of 254 nm with a run time of 15 min. HPLC-grade acetonitrile and water solvents were used. Acetic acid was added as a modifier to achieve a final concentration of 0.1% in each solvent.

Preparative HPLC purification was carried out using an Agilent 1100 HPLC system, RP-C18 column (250 × 10 mm; particle size 10 μm; Luna) with column oven temperature set at 25 °C and a gradient system of eluent water (A) and acetonitrile (B). The gradient condition was as follows: 0–2 min (5% B), 2–5 min (5% B→50% B), 5–10 min (50% B→100% B), 10–15 min (100% B). The flow rate of the solvent was 3.0 mL/min, and the injection volume was 25 μL. All analysis was carried out at wavelength of 254 nm with a run time of 15 min. HPLC-grade acetonitrile and water solvents were used. Acetic acid was added as a modifier to achieve a final concentration of 0.1% in each solvent.

Other common chromatographic techniques, such as thin layer chromatography (TLC) on precoated silica gel G_254_ aluminum plates and silica gel flash column chromatography, were also engaged in the purification of the synthesized compounds.

## 3. Experimental Methods

### 3.1. General Method for Synthesis of (S,E)-11-[2-(Arylmethylene)hydrazono]-pyrrolo [2,1-c] [1,4] Benzodiazepines (***4a***–***4q***)

To a solution of 3 (230 mg, 1.0 mmol) in anhydrous methanol (10 mL) was added aldehyde (1.0 mmol). 3 Å molecular sieves (0.5 g) was also added and stirred at room temperature. Various modifications were used to obtain compounds **4a**–**4q**.(Spectroscopic data for PBD analogs (**4****a**–**4****q**) and typical MRM chromatograms of 4k in plasma and tissues are available in Appendix A)

### 3.2. (S,E)-11-[(1H-Pyrrol-2-yl)methylene)hydrazono]-pyrrolo[2,1-c][1,4] Benzodiazepine (***4a***)

Starting material pyrrole-2-carboxaldehyde (57.0 mg, 0.60 mmol) was used and the reaction mixture was stirred under nitrogen gas overnight for 15 h. Extraction was performed using chloroform/isopropanol (2:1) (3 × 20 mL), where the organic layers were dried over anhydrous Na_2_SO_4_. The solvent mixture was then removed in vacuo and washed with diethyl ether, filtered off, and dried to afford a yellow solid of **4a**. Yield 114.0 mg (62.0%); ^1^H-NMR (400 MHz, DMSO-*d_6_*): δ =1.95 (m, 3H), 2.78 (bd, 1H), 3.56 (m, 2H), 4.43 (bd, 1H, H-11a), 6.17 (s, 1H), 6.51 (s, 1H), 7.10 (s, 1H), 7.19 (m, 2H), 7.53 (t, *J* = 8.0 Hz, 1H), 7.78 (d, *J* = 8.0 Hz, 1H), 8.21 (s, 1H, CH), 9.52 (s, 1H, NH); ^13^C-NMR (100 MHz, DMSO-*d_6_*): δ = 23.4, 26.2, 47.3, 55.3 (C-11a), 109.9, 114.6, 121.9, 122.8, 123.3, 126.5, 128.9, 131.1, 132.6, 138.0, 147.1, 156.9, 165.5 (C=O).

### 3.3. (S,E)-11-[(1-Methyl-1H-imidazol-2-yl)methylene)hydrazono]-pyrrolo[2,1-c][1,4] Benzodiazepine (***4b***)

Starting material 1-methylimidazole-2-carboxaldehyde (95.8 mg, 0.87 mmol) was used and the reaction mixture was stirred under nitrogen gas overnight for 15 h. Extraction was performed using chloroform/isopropanol (2:1) (3 × 20 mL). The combined organic layers were dried over anhydrous Na_2_SO_4_. The solvent mixture was then removed in vacuo and washed with diethyl ether, filtered off, and dried to afford an off-white solid of **4b**. Yield 182.0 mg (65.0%); ^1^H-NMR (400 MHz, DMSO-*d_6_*): δ =1.98 (m, 3H), 2.81 (bd, 1H), 3.56 (m, 2H), 4.03 (s, 3H), 4.43 (bd, 1H, H-11a), 7.09 (s, 1H), 7.17 (m, 1H), 7.33 (d, *J* = 8.0 Hz, 1H), 7.39 (s, 1H), 7.48 (bt, 1H), 7.77 (d, *J* = 8.0 Hz, 1H), 8.30 (s, 1H, CH); ^13^C-NMR (100 MHz, DMSO-*d_6_*): δ = 23.3, 26.5, 35.7, 47.4, 55.3 (C-11a), 122.6, 123.7, 126.2, 126.6, 129.7, 130.8, 131.1, 132.4, 137.6, 142.3, 156.7, 165.3 (C=O).

### 3.4. (S,E)-11-[(Thiazol-5-ylmethylene)hydrazono]-pyrrolo[2,1-c][1,4] Benzodiazepine (***4c***)

Starting material 5-formylthiazole (56.6 mg, 0.60 mmol) was used and the reaction mixture was stirred under nitrogen gas overnight for 15 h. Extraction was performed using chloroform/isopropanol (2:1) (3 × 20 mL), where the organic layers were dried over anhydrous Na_2_SO_4_. The solvent mixture was then removed in vacuo and washed with diethyl ether, filtered off, and dried to afford a yellow solid of **4c**. Yield 115.5 mg (71.0%); ^1^H-NMR (400 MHz, DMSO-*d_6_*): δ =2.08 (m, 3H), 3.05 (bd, 1H), 3.59 (m, 2H), 4.49 (bd, 1H, H-11a), 7.19 (t, *J* = 7.2 Hz, 1H), 7.37 (d, *J* = 8.0 Hz, 1H), 7.48 (t, *J* = 7.2 Hz, 1H), 7.76 (d, *J* = 8.0 Hz, 1H), 8.36 (s, 1H), 8.43 (s, 1H), 9.26 (s, 1H, CH), 9.88 (s, 1H, NH); ^13^C-NMR (100 MHz, DMSO-*d_6_*): δ = 23.7, 26.5, 47.4, 55.8 (C-11a), 122.9, 123.8, 126.9, 127.8, 130.7, 132.4, 137.5, 144.7, 149.2, 158.2, 160.6, 165.3 (C=O).

### 3.5. (S,E)-11-[(2-Methylthiazol-5-yl)methylene)hydrazono]-pyrrolo[2,1-c][1,4] Benzodiazepine (***4d***)

Starting material 2-methylthiazole-5-carboxaldehyde (166.0 mg, 1.305 mmol) was used and the reaction mixture was stirred under nitrogen gas overnight for 15 h. Extraction was performed using chloroform/isopropanol (2:1) (3 × 20 mL), where the organic layers were dried over anhydrous Na_2_SO_4_. The solvent mixture was then removed in vacuo and washed with diethyl ether, filtered off, and dried to afford a yellow solid of **4d**. Yield 319.0 mg (72.0%); ^1^H-NMR (400 MHz, DMSO-*d_6_*): δ =2.01 (m, 3H), 3.04 (bd, 1H), 3.37 (s, 3H), 3.56 (m, 2H), 4.47 (bd, 1H, H-11a), 7.18 (t, *J* = 7.4 Hz, 1H), 7.36 (d, *J* = 7.4 Hz, 1H), 7.47 (t, *J* = 7.4 Hz, 1H), 7.76 (d, *J* = 7.4 Hz, 1H), 8.17 (s, 1H), 8.23 (s, 1H, CH); ^13^C-NMR (100 MHz, DMSO-*d_6_*): δ = 19.3, 23.8, 26.5, 47.4, 55.7 (C-11a), 122.8, 123.8, 126.9, 128.0, 130.8, 132.3, 137.5, 145.0, 148.6, 158.1, 165.2 (C=O), 172.3.

### 3.6. (S,E)-11-[(4-Methylthiazol-5-yl)methylene)hydrazono]-pyrrolo[2,1-c][1,4] Benzodiazepine (***4e***)

Starting material 4-methylthiazole-5-carboxaldehyde (122.0 mg, 0.957 mmol) was used and the reaction mixture was stirred under nitrogen gas overnight for 15 h. Extraction was performed using chloroform/isopropanol (2:1) (3 × 20 mL), where the organic layers were dried over anhydrous Na_2_SO_4_. The solvent mixture was then removed in vacuo and washed with diethyl ether, filtered off, and dried to afford an off-white solid of **4e**. Yield 198.0 mg (61.0%); ^1^H-NMR (400 MHz, DMSO-*d_6_*): δ =2.01 (m, 3H), 3.04 (bd, 1H), 3.37 (s, 3H), 3.56 (m, 2H), 4.47 (bd, 1H, H-11a), 7.18 (m, 1H), 7.36 (m, 1H), 7.47 (m, 1H), 8.25 (s, 1H), 8.67 (s, 1H, CH); ^13^C-NMR (100 MHz, DMSO-*d_6_*): δ = 16.3, 23.5, 26.2, 47.4, 55.8 (C-11a), 122.8, 123.8, 126.9, 128.5, 130.7, 132.3, 137.5, 144.3, 149.2, 155.7, 158.1, 165.3 (C=O).

### 3.7. (S,E)-11-[(2-Aminothiazol-5-yl)methylene)hydrazono]-pyrrolo[2,1-c][1,4] Benzodiazepine (***4f***)

Starting material 2-aminothiazole-5-carboxaldehyde (64.0 mg, 0.50 mmol) was used and the reaction mixture was stirred under nitrogen gas overnight for 15 h. Extraction was performed using chloroform/isopropanol (2:1) (3 × 20 mL), where the organic layers were dried over anhydrous Na_2_SO_4_. The solvent mixture was then removed in vacuo and washed with diethyl ether, filtered off, and dried to afford an off-white solid of **4f**. Yield 115.6 mg (68.0%). ^1^H-NMR (400 MHz, DMSO-*d_6_*): δ =2.05 (m, 3H), 3.03 (bs, 1H), 3.56 (m, 2H), 4.43 (bd, 1H, H-11a), 7.14 (t, *J* = 7.5 Hz, 1H), 7.34 (d, *J* = 7.5 Hz, 1H), 7.45 (d, *J* = 7.5 Hz, 1H), 7.73 (d, *J* = 7.5 Hz, 1H), 7.91 (s, 1H, CH), 9.55 (s, 1H, NH); ^13^C-NMR (100 MHz, DMSO-*d_6_*): δ = 23.6, 26.6, 47.2, 55.7 (C-11a), 117.3, 122.5, 123.3, 126.5, 130.7, 132.3, 138.0, 146.6, 148.6, 156.2, 165.4 (C=O), 175.5.

### 3.8. (S,E)-11-[(2,4-Dichlorothiazol-5-yl)methylene)hydrazono]-pyrrolo[2,1-c][1,4] Benzodiazepine (***4g***)

Starting material 2,4-dichlorothiazole-5-carboxaldehyde (182.0 mg, 1.0 mmol) was used and the reaction mixture was stirred under nitrogen gas overnight for 15 h. Extraction was performed using chloroform/isopropanol (2:1) (3 × 20 mL), where the organic layers were dried over anhydrous Na_2_SO_4_. The solvent mixture was then removed in vacuo and washed with diethyl ether, filtered off, and dried to afford an off-white solid of **4g**. Yield 272.0 mg (69.0%); ^1^H-NMR (400 MHz, DMSO-*d_6_*): δ =2.01 (m, 3H), 2.89 (bd, 1H), 3.58 (m, 2H), 4.50 (bs, 1H, H-11a), 7.21 (s, 1H), 7.39 (s, 1H), 7.48 (bs, 1H), 7.78 (s, 1H), 8.07 (s, 1H, CH), 10.17 (s, 1H, NH); ^13^C-NMR (100 MHz, CDCl_3_): δ = 23.5, 26.8, 47.3, 55.6, 121.9, 123.2, 124.4, 127.2, 130.6, 132.4, 137.1, 139.4, 141.0, 156.6, 160.3, 165.1 (C=O).

### 3.9. (S,E)-11-[2-(Benzo[b]thiophenylmethylene)hydrazono]-pyrrolo[2,1-c][1,4] Benzodiazepine (***4h***)

Starting material benzo[b]thiophene-2-carbaldehyde (100.0 mg, 0.61 mmol) was used and the reaction mixture was stirred under nitrogen gas overnight for 15 h. Extraction was performed using chloroform/isopropanol (2:1) (3 × 20 mL). The combined organic layers were dried over anhydrous Na_2_SO_4_. The solvent mixture was then removed in vacuo and washed with diethyl ether, filtered off, and dried to afford an off-white solid of **4h**. Yield 172.0 mg (75.0%); ^1^H-NMR (400 MHz, CDCl_3_): δ =2.02–2.07 (m, 3H), 2.98–3.02 (m, 1H), 3.68–3.76 (m, 1H), 3.80–3.87 (m, 1H), 4.39 (d, *J* = 7.6 Hz, 1H, H-11a), 7.04 (d, *J* = 7.3 Hz, 1H), 7.22 (t, *J* = 7.3 Hz, 1H), 7.40 (m, 2H), 7.47 (m, 1H), 7.59 (s, 1H), 7.79-7.86 (m, 2H), 8.00 (d, *J* = 7.6 Hz, 1H), 8.53 (s, 1H, CH), 8.71 (s, 1H, NH); ^13^C-NMR (100 MHz, CDCl_3_): δ = 23.5, 25.9, 47.4, 55.4 (C-11a), 120.8, 122.6, 123.9, 124.4, 124.7, 126.2, 126.4, 128.4, 131.4, 132.4, 136.6, 139.4, 139.9, 140.6, 151.6, 157.8, 165.9 (C=O).

### 3.10. (S,E)-11-[(Benzo[b]thiophen-2-ylmethylene)hydrazono]-7-bromo-pyrrolo[2,1-c][1,4] Benzodiazepine (***4i***)

Starting material benzothiophene-2-carboxaldehyde (81.0 mg, 0.50 mmol) was used and the reaction mixture was stirred under nitrogen gas overnight for 15 h. Extraction was performed using chloroform/isopropanol (2:1) (3 × 20 mL), where the organic layers were dried over anhydrous Na_2_SO_4_. The solvent mixture was then removed in vacuo and washed with diethyl ether, filtered off, and dried to afford an off-white solid of **4i**. Yield 158.6 mg (70.0%); ^1^H-NMR (400 MHz, DMSO-*d_6_*): δ =2.00 (m, 3H), 2.78 (bs, 1H), 3.58 (m, 2H), 4.48 (bd, 1H, H-11a), 7.41 (m, 1H), 7.68 (dd, *J* = 2.3, 8.8 Hz, 1H), 7.84 (d, *J* = 2.3 Hz, 1H), 7.89 (m, 2H), 7.98 (bd, 1H), 8.75 (s, 1H, CH), 9.31 (s, 1H, NH); ^13^C-NMR (100 MHz, DMSO-*d_6_*): δ = 23.5, 26.3, 47.4, 55.3 (C-11a), 115.5, 123.1, 125.0, 125.1, 125.3, 126.7, 128.4, 129.4, 132.9, 134.9, 137.1, 139.6, 140.2, 140.5, 152.2, 156.4, 164.0 (C=O).

### 3.11. (S,E)-11-[(Benzo[b]thiophen-3-ylmethylene)hydrazono]-7-bromo-pyrrolo[2,1-c][1,4] Benzodiazepine (***4j***)

Starting material benzothiophene-3-carboxaldehyde (81.0 mg, 0.50 mmol) was used and the reaction mixture was stirred under nitrogen gas overnight for 15 h. Extraction was performed using chloroform/isopropanol (2:1) (3 × 20 mL), where the organic layers were dried over anhydrous Na_2_SO_4_. The solvent mixture was then removed in vacuo and washed with diethyl ether, filtered off, and dried to afford an off-white solid of **4j**. Yield 163.0 mg (72.0%).^1^H-NMR (400 MHz, DMSO-*d_6_*): δ = 1.97 (m, 3H), 2.83 (bs, 1H), 3.59 (m, 2H), 4.53 (bd, 1H, H-11a), 7.38 (d, *J* = 8.6 Hz 1H), 7.49 (m, 2H), 7.68 (d, *J* = 8.6 Hz, 1H), 7.84 (bs, 1H), 8.06 (d, *J* = 7.5 Hz, 1H), 8.46 (s, 1H), 8.75 (bd, 1H), 8.77 (s, 1H, CH), 9.26 (s, 1H, NH); ^13^C-NMR (100 MHz, DMSO-*d_6_*): δ = 23.7, 26.4, 47.6, 55.6 (C-11a), 115.3, 123.3, 125.1, 125.5, 125.6, 125.7, 128.4, 132.1, 132.9, 133.9, 134.9, 136.5, 137.2, 140.4, 152.9, 155.7, 164.1 (C=O).

### 3.12. (S,E)-11-[(Benzo[b]thiophen-2-ylmethylene)hydrazono]-8-chloro-pyrrolo[2,1-c][1,4] Benzodiazepine (***4k***)

Starting material benzothiophene-2-carboxaldehyde (81.0 mg, 0.50 mmol) was used and the reaction mixture was stirred under nitrogen gas overnight for 15 h. Extraction was performed using chloroform/isopropanol (2:1) (3 × 20 mL), where the organic layers were dried over anhydrous Na_2_SO_4_. The solvent mixture was then removed in vacuo and washed with diethyl ether, filtered off, and dried to afford an off-white solid of **4k**. Yield 153.0 mg (75.0%); ^1^H-NMR (400 MHz, DMSO-*d_6_*): δ = 1.99 (m, 3H), 2.79 (bs, 1H), 3.58 (m, 2H), 4.49 (bd, 1H, H-11a), 7.23 (d, *J* = 8.5 Hz, 1H), 7.42 (m, 2H), 7.60 (bs, 1H), 7.77 (d, *J* = 8.5 Hz, 1H), 7.94 (m, 3H), 8.75 (s, 1H, CH), 9.34 (s, 1H, NH); ^13^C-NMR (100 MHz, DMSO-*d_6_*): δ = 23.3, 26.3, 47.4, 55.3 (C-11a), 122.1, 123.1, 123.5, 125.0, 125.2, 125.4, 126.6, 129.4, 132.8, 136.5, 138.9, 139.6, 140.3, 140.6, 152.3, 156.3, 164.5 (C=O).

### 3.13. (S,E)-11-[(3-Methylbenzo[b]thiophen-2-yl)methylene)hydrazono]-pyrrolo[2,1-c][1,4] Benzodiazepine (***4l***)

Starting material 3-methylbenzo[b]thiophene-2-carboxaldehyde (106.0 mg, 0.60 mmol) was used and the reaction mixture was stirred under nitrogen gas overnight for 15 h. Extraction was performed using chloroform/isopropanol (2:1) (3 × 20 mL), where the organic layers were dried over anhydrous Na_2_SO_4_. The solvent mixture was then removed in vacuo and washed with diethyl ether, filtered off, and dried to afford a yellow solid of **4l**. Yield 152.0 mg (65.0%); ^1^H-NMR (400 MHz, DMSO-*d_6_*): δ = 2.00 (m, 3H), 2.82 (bs, 1H), 3.37 (s, 3H), 3.58 (m, 2H), 4.44 (bd, 1H, H-11a), 7.19 (t, *J* = 7.0 Hz, 1H), 7.44 (m, 4H), 7.84 (m, 3H), 8.85 (s, 1H, CH), 9.01 (s, 1H, NH); ^13^C-NMR (100 MHz, DMSO-*d_6_*): δ = 11.8, 23.5, 26.3, 47.2, 55.3 (C-11a), 122.5, 122.9, 123.1, 123.7, 124.9, 126.6, 126.9, 130.8, 132.5, 134.4, 136.4, 137.4, 139.9, 140.6, 150.6, 156.9, 165.3 (C=O).

### 3.14. (S,E)-11-[(3-Chlorobenzo[b]thiophen-2-yl)methylene)hydrazono]-pyrrolo[2,1-c][1,4] Benzodiazepine (***4m***)

Starting material 3-chloro-1-benzothiophene-2-carboxaldehyde (98.4 mg, 0.60 mmol) was used and the reaction mixture was stirred under nitrogen gas overnight for 15 h. Extraction was performed using chloroform/isopropanol (2:1) (3 × 20 mL), where the organic layers were dried over anhydrous Na_2_SO_4_. The solvent mixture was then removed in vacuo and washed with diethyl ether, filtered off, and dried to afford an off-white solid of **4m**. Yield 122.7 mg (60.0%); ^1^H-NMR (400 MHz, DMSO-*d_6_*): δ =2.12 (m, 3H), 3.16 (bs, 1H), 3.61 (m, 2H),.4.52 (bd, 1H, H-11a), 7.21 (t, *J* = 7.2 Hz, 1H), 7.39 (m, 1H), 7.54 (m, 2H), 7.78 (d, *J* = 7.2 Hz, 1H), 7.89 (m, 1H), 8.07 (m, 1H), 8.39 (s, 1H, CH), 9.98 (s, 1H, NH); ^13^C-NMR (100 MHz, DMSO-*d_6_*): δ = 23.8, 26.9, 47.4, 55.8 (C-11a), 122.5, 122.9, 123.4, 124.0, 124.3, 125.9, 126.2, 127.0, 128.1, 130.8, 132.5, 134.6, 137.4, 140.8, 142.9, 158.9, 165.2 (C=O).

### 3.15. (S,E)-11-[(Benzo[d]thiazol-2-ylmethylene)hydrazono]-pyrrolo[2,1-c][1,4] Benzodiazepine (***4n***)

Starting material 1,3-benzothiazole-2-carboxaldehyde (142.0 mg, 0.87 mmol) was used and the reaction mixture was stirred under nitrogen gas overnight for 15 h. Extraction was performed using chloroform/isopropanol (2:1) (3 × 20 mL), where the organic layers were dried over anhydrous Na_2_SO_4_. The solvent mixture was then removed in vacuo and washed with diethyl ether, filtered off, and dried to afford a yellow solid of **4n**. Yield 222.0 mg (68.0%); ^1^H-NMR (400 MHz, DMSO-*d_6_*): δ =1.97 (m, 3H), 2.80 (bd, 1H), 3.57 (m, 2H), 4.50 (bd, 1H, H-11a), 7.40 (m, 3H), 7.68 (dd, *J* = 8.9, 2.2 Hz, 1H), 7.84 (d, *J* = 2.2 Hz, 1H), 7.90 (m, 2H), 7.98 (bd, 1H), 8.75 (s, 1H, CH), 9.30 (s, 1H, NH); ^13^C-NMR (100 MHz, DMSO-*d_6_*): δ = 23.4, 26.3, 47.4, 55.3 (C-11a), 115.4, 123.1, 124.9, 124.4, 125.2, 126.7, 128.5, 129.4, 132.9, 134.9, 137.0, 139.6, 140.3, 140.6, 152.1, 156.4, 164.0 (C=O).

### 3.16. (S,E)-11-[(1-Methyl-1H-indol-3-yl)methylene)hydrazono]-pyrrolo[2,1-c][1,4] Benzodiazepine (***4o***)

Starting material 1-methylindole-3-carboxaldehyde (152.0 mg, 0.957 mmol) was used and the reaction mixture was stirred under nitrogen gas overnight for 15 h. The combined organic layers were dried over anhydrous Na_2_SO_4_. The solvent mixture was then removed in vacuo and washed with diethyl ether, filtered off, and dried to afford a yellow solid of **4o**. Yield 244.0 mg (69.0%); ^1^H-NMR (400 MHz, DMSO-*d_6_*): δ =1.98 (m, 3H), 2.87 (bs, 1H), 3.36 (s, 3H), 3.58 (m, 2H), 4.45 (bs, 1H, H-11a), 7.23 (m, 3H), 7.34 (bd, 1H), 7.51 (m, 2H), 7.78 (d, *J* = 8.0 Hz, 1H), 7.92 (s, 1H), 8.38 (d, *J* = 8.0 Hz, 1H), 8.65 (s, 1H, CH), 8.98 (s, 1H, NH); ^13^C-NMR (100 MHz, DMSO-*d_6_*): δ = 23.5, 26.2, 33.3, 47.3, 55.5 (C-11a), 55.6, 110.6, 111.7, 121.4, 122.3, 122.9, 123.1, 125.6, 126.4, 130.9, 132.4, 132.5, 135.8, 138.0, 138.1, 153.3, 154.7, 165.5 (C=O).

### 3.17. (S,E)-11-[(5-Methoxy-1H-indol-3-yl)methylene)hydrazono]-pyrrolo[2,1-c][1,4] Benzodiazepine (***4p***)

Starting material 5-methoxy-1-methylindole-3-carbaldehyde (100.0 mg, 0.57 mmol) was used and the reaction mixture was stirred under nitrogen gas overnight for 15 h. Extraction was performed using chloroform/isopropanol (2:1) (3 × 20 mL). The combined organic layers were dried over anhydrous Na_2_SO_4_. The solvent mixture was then removed in vacuo and washed with diethyl ether, filtered off, and dried to afford a brown solid of **4p**. Yield 153.0 mg (68.0%); ^1^H-NMR (400 MHz, DMSO-*d_6_*): δ =1.99 (m, 3H), 2.84 (bs, 1H), 3.37 (s, 3H), 3.60 (m, 2H), 4.47 (bs, 1H, H-11a), 6.84 (bd, 1H), 7.14 (bt, 1H), 7.34 (m, 2H), 7.49 (bt, 1H), 7.77 (bd, 1H), 7.80 (d, *J* = 2.4 Hz, 1H), 7.89 (d, *J* = 2.4 Hz, 1H), 8.67 (s, 1H, CH), 9.01 (s, 1H, NH); ^13^C-NMR (100 MHz, DMSO-*d_6_*): δ = 23.6, 26.2, 47.4, 55.4 (C-11a), 55.6, 104.0, 112.4, 112.9, 113.0, 121.7, 123.0, 125.8, 126.0, 131.0, 132.3, 132.4, 132.6, 138.1, 153.7, 154.6, 154.9, 165.5 (C=O).

### 3.18. (S,E)-11-[(Adamantan-1-yl)methylene)hydrazono]-pyrrolo[2,1-c][1,4] Benzodiazepine (***4q***)

Starting material adamantane-1-carboxaldehyde (164.0 mg, 1.0 mmol) was used and the reaction mixture was stirred under nitrogen gas overnight for 15 h. Extraction was performed using chloroform/isopropanol (2:1) (3 × 20 mL), where the organic layers were dried over anhydrous Na_2_SO_4_. The solvent mixture was then removed in vacuo and washed with diethyl ether, filtered off, and dried to afford a yellow solid of **4q**. Yield 256.0 mg (68.0%); ^1^H-NMR (400 MHz, CDCl_3_): δ =1.74 (m, 11H), 2.01 (m, 7H), 2.91 (bd, 1H), 3.75 (m, 2H), 4.29 (bd, 1H, H-11a), 6.95 (d, *J* = 9.5 Hz, 1H), 7.15 (t, *J* = 8.4 Hz, 1H), 7.41 (t, *J* = 7.4 Hz, 1H), 7.95 (d, *J* = 8.4 Hz, 1H), 8.40 (s, 1H, CH); ^13^C-NMR (100 MHz, CDCl_3_): δ = 23.4, 26.0, 27.9, 36.7, 37.3, 39.7, 47.3, 50.6, 55.3 (C-11a), 120.6, 123.5, 126.1, 131.3, 132.3, 136.9, 156.9, 166.1, 168.9 (C=O).

## 4. Cannabinoid Receptor Binding Assay

The affinities of the compounds for CB1 and CB2 receptors were examined using displacement assays, as previously described [13]. Briefly, cell membranes from CHO cells expressing human CB1 or human CB2 receptors were isolated using differential centrifugation. Test compounds reconstituted in DMSO and were incubated with the isolated membrane in binding buffer (50 mM Tris-HCl, 1 mM EDTA, 3 mM MgCl_2_, 5 mg/mL BSA, pH 7.4) along with 2.5 nM [^3^H]CP-55,940. Total binding was assessed in the presence of equal concentration of DMSO, while non-specific binding was determined in the presence of 10 μM CP-55,940, and background binding was determined in wells lacking membrane. Following incubation at 30 °C for 60 min, the binding reactions were terminated by vacuum filtration through Whatman GF/C filters. The filters were then washed twice with ice-cold buffer (50 mM Tris-HCl, 1 mg/mL BSA). Liquid scintillation cocktail was added to each well and the total tritiated counts per minute were analyzed using a TopCount scintillation counter. Background counts were subtracted from all wells and the percent displacement from total binding was calculated.

The compounds were initially screened at 10 μM concentrations. If they produced at least ±30% displacement of the radioligand, then full competition curves were constructed. *K_i_* values were calculated using GraphPad Prism (San Diego, CA, USA) and K_d_ values determined using a 1 site fit. All assays were run in technical and biological replicates so that the n = 5–6.

## 5. PK Evaluation in CD-1 Mice

### 5.1. Preparation of Stock Solutions, Calibration Standards, Quality Control Samples and Internal Standard Solution

The primary stock solutions of **4k** and phenacetin (internal standard, IS) were prepared in methanol at a concentration of 1.0 mg/mL. Working solutions of calibration standards and quality control (QC) samples were prepared by dilution with methanol and stored at −20 °C. A working stock of the IS solution (20 ng/mL) was prepared in methanol and stored at −20 °C.

### 5.2. Instrument and Analytical Conditions

Chromatography was performed on an Acquity^TM^ UPLC system (Waters Corp, Milford, MA, USA) with an autosampler temperature at 10 °C. Waters Acquity UPLC^®^ HSS C18 column (3.0 × 50 mm, 1.8 µm particle size) was used for chromatographic separation with linear gradient elution consisting of (A) 90% acetonitrile and (B) 10% 0.2% formic acid in Milli-Q water as mobile phases. The flow rate was set at 0.30 mL/min, and the injection volume was 2 µL.

An Acquity Tandem Quadrupole Mass Detector (Xevo TQ-S; Waters Corp, Milford, MA, USA) in positive electrospray ionization mode was used for mass spectrometric detection. For collision-induced dissociation, argon was used as collision gas. The cone voltage and collision energy were set at 60 V and 34 V for **4k** and 46 V and 26 V for the IS, respectively. Quantification was performed using the monitoring of multiple reaction monitoring (MRM) of following transitions: *m*/*z* 409.0/220.9 for **4k** and *m*/*z* 180.0/92.7 for IS. Retention times of **4k** and IS were 2.85 and 1.81 min, respectively.

### 5.3. Sample Preparation

A simple protein precipitation method was followed for extraction of **4k** from mice plasma. To an aliquot of 50 μL of plasma or tissue (brain, liver, or kidney) samples, IS solution (5 μL of 20 ng/mL) was added and mixed for 15 s on a cyclomixer (Thermo Scientific, Indianapolis, IN, USA). After precipitation with 200 μL of acetonitrile, the mixture was vortexed for 2 min, followed by centrifugation for 10 min at 14,000 rpm on an accuSpin Micro 17R (Fisher Scientific, Suwanee, GA, USA) at 5 °C. An aliquot of ~150 μL of clear supernatant was transferred into vials and 2 μL was injected onto LC-MS/MS system for analysis.

### 5.4. In Vivo Studies in CD-1 Mice

All work involving animal subjects was pre-approved by the University of Mississippi Institutional Animal Care and Use Committee and was conducted in accordance with ethical guidelines defined by the National Institutes of Health (NIH Publication No. 85-23).

### 5.5. Subjects

Male CD-1 mice (N = 48) were obtained from Envigo (St. Louis, MO, USA) and maintained on a 12:12 h reversed dark/light cycle (lights off at 07:00 h) with ad libitum access to food and water. Following a ~4 h fast, animals were randomly assigned to one of two groups. Group I and II animals (n = 24/group, weight range 25–30 g) received compound **4k** (5 mg/kg) orally (in the form of a solution, prepared using 10% absolute alcohol, 10% cremophor, and 80% Milli-Q water) or intravenously (*i.v.*, using solution formulation comprising 10% absolute alcohol, 10% cremophor, and 80% normal saline), respectively. At 0.5, 1, 2, 4, 8, and 24 h post administration, animals were euthanized, blood was collected in K_2_. EDTA-containing polypropylene tubes and, tissues were collected (brain, liver, and kidney). Plasma was harvested by centrifuging the blood using Eppendorf 5430R Centrifuge (Germany) at 5000 rpm for 5 min and stored frozen at −80 °C until analysis.

### 5.6. Tissue Preparation

Brain, liver, and kidney tissues were homogenized in separate 15 mL round-bottom screw-capped vials in phosphate buffered saline (5 volumes of each tissue mass) with a homogenizer (Polytron^®^) and stored at −80 °C until analysis. Plasma or tissue homogenates (50 μL) were spiked with IS and processed as mentioned in sample preparation section.

### 5.7. Pharmacokinetic Assessments

Plasma and tissue concentration and time data of compound **4k** were analyzed by a non-compartmental method using WinNonlin Version 5.3 (Pharsight Corporation, Mountain View, CA, USA).

## 6. Results and Discussion

### 6.1. Chemistry

To gain a better understanding of the structure–activity relationships of **4**, we extended the range of PBD-11-hydrazinyl derivatives (**4a**–**4q**) by alkenylation of the primary amine group of the cycloamidine **3** via the formation of a Schiff’s base with various carboxaldehydes [13]. Synthetic approaches were initiated based on the stepwise synthesis of dilactam **1** by the cyclocondensation of an equimolar mixture of L-proline and readily available basic structure of PBD natural product, isatoic anhydride (or substituted compound) in DMF at 155 °C. Thionation of compound **1** with 0.5 eq. of 2,4-bis-(4-methoxyphenyl)-1,3-dithia-2,4-diphosphetane-2,4-disulfide (Lawesson’s reagent) in THF at room temperature produced thiolactam **2**. The subsequent treatment of **2** with 98% hydrazine monohydrate in ethanol at room temperature generated cycloamindine **3**. The title compound **3** was further subjected to condensation with several aldehydes in anhydrous MeOH and molecular sieves (3 Å) at room temperature to afford a cluster of highly conjugated Schiff’s base **4** in high yield. This procedure involves the initial formation of the intermediate carbinol amine, which dehydrates to form an imine. The reaction time was varied from a few minutes to a couple of hours and gave the desired products in good yields. The alkenylation of **3** with aldehydes was best performed under optimized conditions in which the parent molecule was exposed to five equivalents of aldehyde in methanol for varied reaction times at room temperature. Upon further crystallization of crude product from EtOAc/hexanes, pure (*S*,*E*)-11-[2-(arylmethylene)hydrazono]-PBD analogs (**4a**–**4q**) were formed as crystalline solids. The chemical structures are shown in Figure 2. These compounds are structurally unique from any reported small-molecule CB ligands, confirming the chemotype novelty of all the generated compounds.

### 6.2. Biological Evaluation of the Synthesized Compounds

#### 6.2.1. Cannabinoid Receptors Displacement Assay

The CB1/CB2 receptor binding activities of analogs **4a**–**4q** were assessed via radioactivity-based competitive binding assays using [^3^H]CP55,940, an extensively studied radioligand that is frequently used for cannabinoid assays [14,15,16]. The highly potent and nonselective CB agonist CP55,940 binds to the same orthosteric active site where known CB agonists are known to bind. In preliminary screening, all 17 synthetic PBD compounds (**4a**–**4q**) were subjected to in vitro CB1 and CB2 binding assays at a single concentration of 10 µM. The observed percentage displacement (%) of the radioligand at the CB receptors of these analogs are summarized in Table 1. From the 17 compounds evaluated in the competitive radioligand assay, compounds **4h**, **4j**–**4l**, and **4q** showed significant displacement (more than 60%). Among them, two structurally distinct PBD analogs (**4k** and **4q**) displayed the greatest [^3^H]CP-55,940 displacement and selectivity for CB2 over CB1. Therefore, the most promising ligands (**4k** and **4q)** were selected for assessment in full competition curves against CB1 and CB2. The binding assays revealed selective binding affinity of **4k** and **4q** with *K*_i_ values of 146 and 137 nM, respectively, toward CB2 receptors (Table 2).

#### 6.2.2. In Vivo PK Studies of the Lead Analog **4k**

Drug discovery research programs routinely execute mouse PK studies to characterize the PK properties of compounds as a filtering tool to advance drug candidates through the pipeline as well as to support efficacy and toxicology studies. Stability studies on two selected analogs has revealed a better tolerance of **4k** under physiological pH when compared to **4q** compound (not reported). Consequently, **4k** was selected for pharmacokinetic studies in CD1 male mice at 5 mg/kg body weight (Table 3, Figure 3).

In plasma, the concentrations of compound **4k** decreased in a mono-exponential manner after 5.0 mg/kg intravenous administration. The mean clearance (CL) was found to be 23.5 ± 5.35 mL/min/kg, which is 26% of hepatic blood flow in mice. Compound **4k** had a high volume of distribution of 14.3 ± 3.11 L/kg. The terminal half-life (t_1/2_) was found to be 6.91 ± 0.15 h. The post oral administration, maximum plasma concentrations (C_max_: 198 ± 11.51 ng/mL) were attained at 0.88 ± 0.13 h (T_max_), indicating rapid absorption from the gastrointestinal tract. The apparent half-life (16.4 ± 1.66 h) determined after oral administration was longer than that after intravenous administration (6.91 h), which may indicate multiple sites of absorption. The AUC_0–__∞_ attained post oral dose was 739 ± 6.75 ng × h/mL. The oral bioavailability in mice at 5 mg/kg was 18 ± 2.43% (Table 3).

In the brain, the concentrations of compound **4k** decreased mono-exponentially after 5.0 mg/kg intravenous administration. The mean brain clearance (CL) was found to be 15.7 ± 2.31 mL/min/kg. Compound **4k** had a high volume of distribution of 8.82 ± 2.43 L/kg in the brain. The terminal half-life (t_1/2_) was found to be 6.54 ± 1.38 h. Post oral administration, the maximum brain concentrations (C_max_: 602 ± 44.75 ng/g) were attained at 1.25 ± 0.44 h (T_max_). The apparent half-life was found to be 22.9 ± 3.73 h. The AUC_0–__∞_ attained post oral dose was 3069 ± 64 ng × h/g. The apparent oral bioavailability to the brain, as compared to IV administration, was 42.5 ± 5.75% (Table 1). The mean brain to plasma ratios after intravenous and oral administration were 3.67 and 4.61, respectively, which indicates that the compound had very good brain penetration.

In the kidney, the concentrations of compound **4k** decreased mono-exponentially after 5.0 mg/kg intravenous administration. The mean kidney clearance (CL) was found to be 6.38 ± 0.49 mL/min/kg. Compound **4k** had a high volume of distribution of 5.63 ± 0.37 L/kg in the kidney. The terminal half-life (t_1/2_) was found to be 10.2 ± 0.49 h. Post oral administration, the maximum kidney concentrations (C_max_: 795 ± 22.65 ng/g) were attained at 0.75 ± 0.15 h (T_max_). The apparent half-life was found to be 16.7 ± 1.61 h. The AUC_0-__∞_ attained post oral dose was 5223 ± 44.75 ng × h/g. The apparent oral bioavailability in the kidney at 5 mg/kg was 37 ± 3.78% (Table 1). The mean kidney to plasma ratios after intravenous and oral administration were 14.5 and 7.25, respectively.

In the liver, the mean liver clearance (CL) was found to be 4.28 ± 0.38 mL/min/kg. Compound **4k** had a high volume of distribution of 3.38 ± 0.18 L/kg in liver. The terminal half-life (t_1/2_) was found to be 9.26 ± 0.67 h. Post oral administration, the maximum liver concentrations (C_max_:1891 ± 282 ng/g) were attained at 0.75 ± 0.15 h (T_max_). The apparent half-life was found to be 13.0 ± 0.97 h. The AUC_0-__∞_ attained post oral dose was 6423 ± 571 ng × h/g. The apparent oral bioavailability in the liver at 5 mg/kg was 32 ± 2.86% (Table 1). The mean liver to plasma ratios after intravenous and oral administration were 11.5 and 9.89, respectively.

## 7. Conclusions

Through a structure-based rational drug design, we synthesized a subset of 17 (*S*,*E*)-11-[2-(arylmethylene)hydrazono]-PBD derivatives using a multi-step synthesis approach to establish clusters of highly potent and selective CB2 ligands. Most of the designed analogs exhibited a high percentage displacement (%) of the CB1 and CB2 receptor. Most importantly, the most promising compounds, **4k** and **4q,** displayed sub-micromolar efficacy (*K*_i_ of 146 and 137 nM) when tested for in a binding affinity assay.

Our previously reported studies have shown suitable drug-like properties of this class of ligands via the computational calculation of ADMET and physicochemical properties. The reported data validated that almost all the designed analogs possess Abs_risk, CYP_Risk, TOX_Risk, and ADMET_Risk within the satisfactory defined limits [13]. The current pharmacokinetics assessments further support the favorable drug-like behavior of this class in that the oral bioavailability is reasonable and tissue uptake, especially brain penetration, is high, with a prolonged half-life. This suggests the feasibility of central nervous system activity with simple dose regimens. The structure–activity relationships (SAR) suggested that attachment of S-heterocyclic aldehydes (e.g., benzothiophene) to the hydrazine portion, in addition to an 8-chloro-substituted PBD, improves the selectivity of CB2 over CB1. Moreover, the adamantyl pharmacophore establishes extensive hydrophobic interactions with three phenylalanine residues and one histidine residue within the active site of the CB2 receptor [17]. Most notably, these bioactive compounds represent structurally new chemotypes in the area of cannabinoid research and could be considered for further structural optimization as selective CB2 ligands.

## Figures and Tables

**Figure 1 molecules-27-00509-f001:**
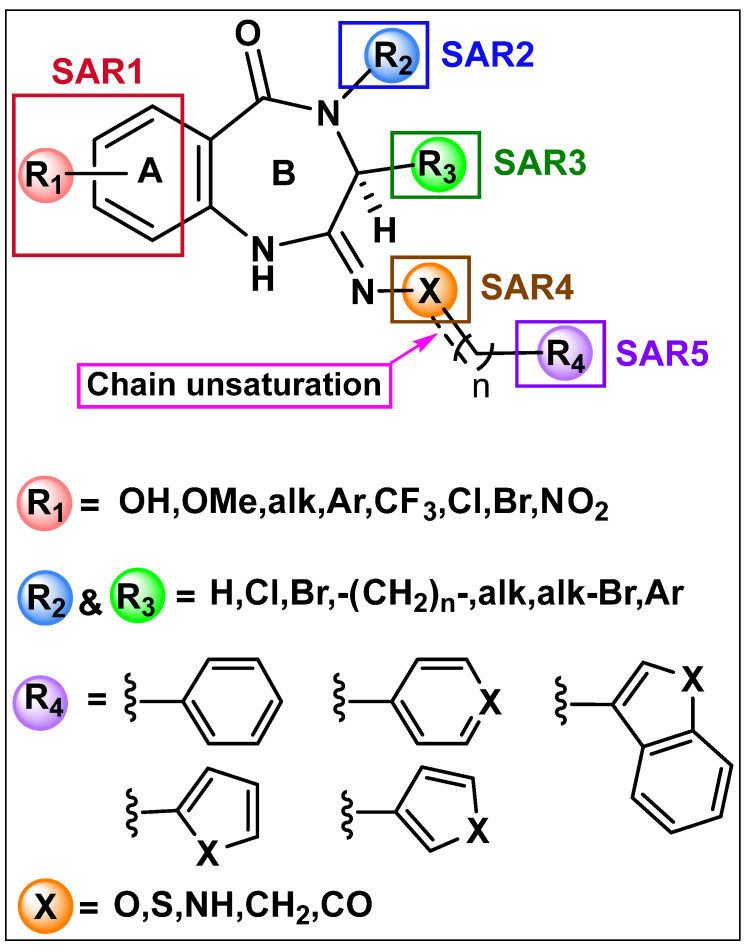
Structures of the proposed analogs.

**Figure 2 molecules-27-00509-f002:**
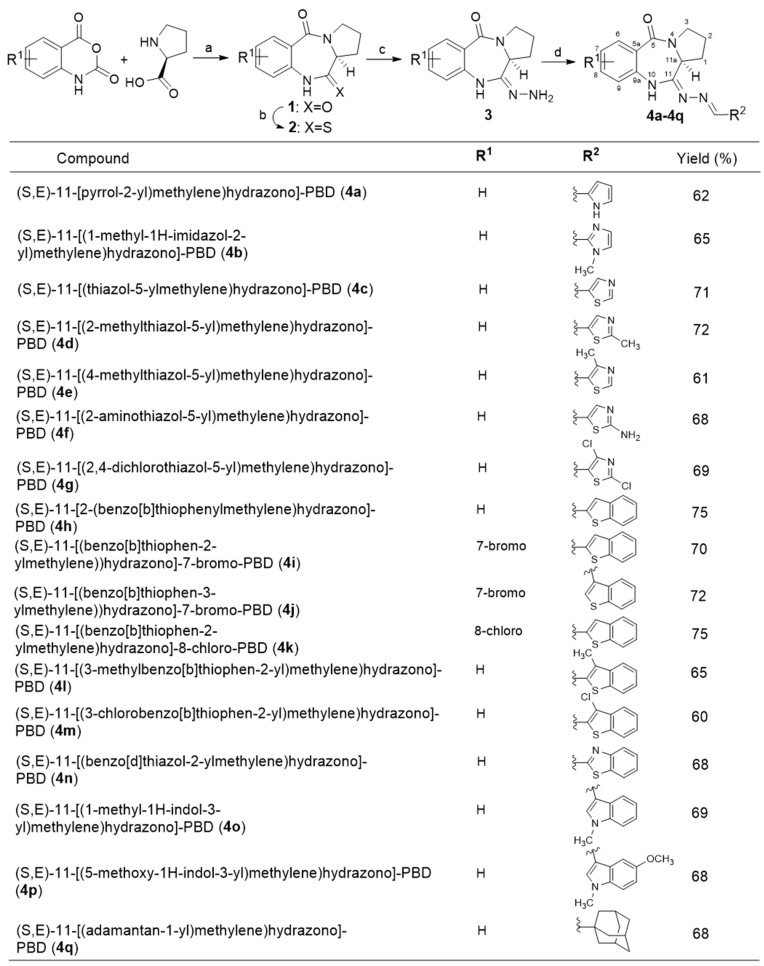
Reagents and conditions: (**a**) DMF, 155 °C, 5 h, 82.0%; (**b**) Lawesson’s reagent, THF, rt, 15 h, 87.0%; (**c**) N_2_H_4_·H_2_O (98%), EtOH(abs.), rt, 15 h, 99.0%; (**d**) Aldehydes, MeOH (anhy.), rt, 15 h, 60–75.0%.

**Figure 3 molecules-27-00509-f003:**
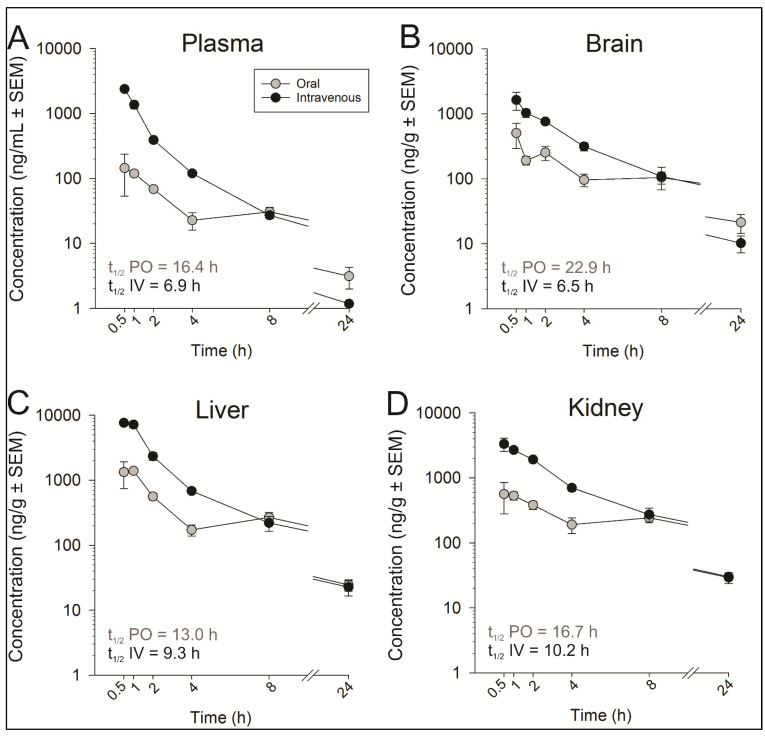
Pharmacokinetic profiles of compound **4k** in plasma (**A**), brain (**B**), liver (**C**), and kidney (**D**) of CD1 mice following intravenous and oral dosing of **4k** to mice at 5 mg/kg.

**Table 1 molecules-27-00509-t001:** Displacement of [^3^H]CP-55,940 from CB1 and CB2 receptors by 10 µM concentration of compounds **4a**–**4q**.

Compound	CB1(%)	CB2(%)	Compound	CB1(%)	CB2(%)
**4a**	−3.09	−12.50	**4j**	66.81	65.24
**4b**	−13.99	−14.48	**4k**	41.26	84.60
**4c**	0.25	3.06	**4l**	77.17	51.73
**4d**	−27.99	−33.65	**4m**	5.22	50.05
**4e**	21.88	19.86	**4n**	44.14	43.34
**4f**	1.75	3.72	**4o**	5.09	1.01
**4g**	24.19	29.27	**4p**	−27.20	−29.47
**4h**	36.88	61.05	**4q**	55.27	95.58
**4i**	49.82	36.39			

**Table 2 molecules-27-00509-t002:** Cannabinoid receptor binding assays of the lead compounds against CB2.

Compound	Structure	*K*_i_ (nM)
**4k**	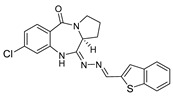	146
**4q**	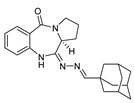	137

**Table 3 molecules-27-00509-t003:** Pharmacokinetic parameters of compound **4k** in CD1 mice following intravenous and oral dosing of **4k** in mice at 5 mg/kg.

PK Parameters		Intravenous				Oral			
		Plasma	Brain	Kidney	Liver	Plasma	Brain	Kidney	Liver
t_1/2_	h	6.91	6.54	10.2	9.26	16.4	22.9	16.7	13.0
C_max_ or C_0_	ng/mL or ng/g	4345	2260	4725	13497	198	602	795	1891
T_max_	h	-	-	-	-	0.88	1.25	0.75	0.75
AUC_0–__∞_	ng × h/mL or ng × h/g	4015	5685	13272	19991	739	3069	5223	6423
CL	mL/min/kg	23.5	15.7	6.38	4.28	-	-	-	-
V_d_	L/kg	14.3	-	-	-	-	-	-	-
F	%	-	-	-	-	18	43	37	32

t_1/2_: half-life; C_max_: maximum concentration; C_0_: plasma concentration estimated at time 0 for IV; T_max_: time at maximum concentration at 0.5 h; AUC_0–__∞_: area under the curve to infinite time; CL: clearance; V_d_: volume of distribution; F: bioavailability.

## Data Availability

Spectroscopic data for PBD analogs (**4a**–**4q**) and MRM chromatograms of **4k** in plasma and tissues are available in the Appendix A.

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
