# Peer review of "Identification of an Orally Bioavailable, Brain-Penetrant Compound with Selectivity for the Cannabinoid Type 2 Receptor"

_molecules, 2022, doi:10.3390/molecules27020509_

Round 1
Reviewer 1 Report
In the study in which 19 (S,E)-11-[2-(arylmethylene)hydrazono]-PBD analogues were synthesized, 4k and 4q emerged as CB2 ligands. Also, the study was evaluated 4Pharmacokinetic profiles of compound 4k in plasma, kidney, brain and liver. The researchers shared the following important data in their results: reasonable bioavailability, high tissue penetration, and a long half-life. All results supported the positive drug-like behavior of this class. I would like to emphasize that this study, which was designed very well by the researchers, is also remarkable in its results. I would like to inform you that I have found it appropriate to publish this article in the molecules journal. Below are some of my small suggestions. The abbreviations of some units should be edited. The unit should be written with ‘kg’ not ‘Kg’ and ‘s’ not ‘sec’.
|

Author Response
Reviewer 1:
Comment 1: In the study in which 19 (S,E)-11-[2-(arylmethylene)hydrazono]-PBD analogues were synthesized, 4k and 4q emerged as CB2 ligands. Also, the study was evaluated 4Pharmacokinetic profiles of compound 4k in plasma, kidney, brain and liver. The researchers shared the following important data in their results: reasonable bioavailability, high tissue penetration, and a long half-life. All results supported the positive drug-like behavior of this class. I would like to emphasize that this study, which was designed very well by the researchers, is also remarkable in its results. I would like to inform you that I have found it appropriate to publish this article in the molecules journal.
Response 1: We appreciate the reviewer’s positive comment.
Comment 2: The abbreviations of some units should be edited. Unit should be written with ‘kg’ not ‘Kg’ and ‘s’ not ‘sec.
Response 1: The units’ abbreviations have been revised as suggested.
Reviewer 2 Report
This manuscript describes work on the synthetization of 19 PBD compounds and characterization of their interactions with CB1 and CB2 receptors. Compounds 4k and 4q were first identified as potent and selective CB2 ligands. In addition, the authors showed that compound 4k is orally-bioavailable and brain-penetrant and exerts favorable pharmacokinetic property. There are some questions need to be clarified by the authors before publishing this manuscript as an article.
- Page 12, 1st paragraph, “Consequently, 4k was selected for dose response studies in CD1 male mice at 5 mg/kg body weight”. The words “dose response” can be replaced with “pharmacokinetic” as only one dose (5 mg/kg) was evaluated in the PK studies.
- Page 12, 1st paragraph, delete “Group I and II animals ……. using WinNonlin Version 5.3”. This information has been provided in Material and Methods.
- Please add representative LC-MS/MS chromatograms of 4k (plasma and tissues) to the supplemental materials.
- Importantly, Vd can be estimated by using plasma concentration-time profile, while it cannot be calculated according to data observed in other tissues such as brain, liver, and kidney. Please delete tissues’ Vd data in Table 3 and revise the descriptions of Vd in the text.
- Page 14, 1st paragraph of Conclusion, “…compounds 4k and 4q were displayed…”, delete “were”.
- Tmax values shown in Table 3 may be inaccurate because maximum concentrations were observed at the first time point (0.5 h) in plasma and tissues (Figure 3).
- Figure 3B-D, the unit for concentration should be “ng/g” ?
Author Response
Reviewer 2:
This manuscript describes work on the synthetization of 19 PBD compounds and characterization of their interactions with CB1 and CB2 receptors. Compounds 4k and 4q were first identified as potent and selective CB2 ligands. In addition, the authors showed that compound 4k is orally-bioavailable and brain-penetrant and exerts favorable pharmacokinetic property. There are some questions need to be clarified by the authors before publishing this manuscript as an article.
Comment 1: Page 12, 1st paragraph, “Consequently, 4k was selected for dose response studies in CD1 male mice at 5 mg/kg body weight”. The words “dose response” can be replaced with “pharmacokinetic” as only one dose (5 mg/kg) was evaluated in the PK studies.
Response 1: The words “dose response” have been replaced with “pharmacokinetic” as suggested.
Comment 2: Page 12, 1st paragraph, delete “Group I and II animals ……. using WinNonlin Version 5.3”. This information has been provided in Material and Methods.
Response 2: “Group I and II animals ……. using WinNonlin Version 5.3” has been deleted as suggested
Comment 3: Please add representative LC-MS/MS chromatograms of 4k (plasma and tissues) to the supplemental materials.
Response 3: Typical MRM chromatograms of 4k in plasma and tissues have been implemented in the supporting information.
Comment 4: Importantly, Vd can be estimated by using plasma concentration-time profile, while it cannot be calculated according to data observed in other tissues such as brain, liver, and kidney. Please delete tissues’ Vd data in Table 3 and revise the descriptions of Vd in the text.
Response 4: The Vd data for the tissues in table 3 has been deleted as suggested.
Comment 5: Page 14, 1st paragraph of Conclusion, “…compounds 4k and 4q were displayed…”, delete “were”.
Response 5: “Were” has been deleted.
Comment 6: Tmax values shown in Table 3 may be inaccurate because maximum concentrations were observed at the first time point (0.5 h) in plasma and tissues (Figure 3).
Response 6: We valued the reviewer’s point of view and we have now clearly stated under table 3 that Tmax has been calculated at 0.5 h.
Comment 7: Figure 3B-D, the unit for concentration should be “ng/g” ?
Response 7: The units in figures 3B-D have been adjusted as suggested
Reviewer 3 Report
The manuscript entitled “Identification of an Orally-Bioavailable, Brain-Penetrant Compound with Selectivity for the Cannabinoid Type 2 Receptor” by Ospanov and coworkers describes synthesis of 19 pyrrolo[2,1-c][1,4]benzodiazepines (PBD) that were designed using a SAR approach in order to produce PBD derivatives selective for CB2 subtype of cannabinoid receptors. These 19 molecules were tested for their selectivity in binding to CB1 and CB2 orthosteric sites using “displacement of [3H]CP-55,940 binding” assay. Two of the compounds showing the greatest CB2 [3H]CP-55,940 binding displacement and CB2 selectivity (4k and 4q) were analysed further. Both molecules displaced radioligand binding to CB2 subtype with a Ki values of about 100 nM. 4k was selected for pharmacokinetic studies in mice due to it’s higher stability under physiological pH. 4k showed good bioavailability and brain penetration, 5 mg/Kg (p.o.) dose producing maximally a brain concentration of 0.6 µg/g (corresponding roughly 1 µM concentration).
The synthesized compounds are all novel CB receptor ligands that have not been studied earlier. Results are interpreted properly. The main findings are significant, because 4k showed relatively high CB2 selectivity and affinity, and good brain penetration producing quite high concentration in the brain. Conclusions are clear and justified and quite directly derived from the results. The manuscript does not include much speculation, because Discussion has been included in Results section and it actually does not contain much discussion outside Results. The research is at the stage where some promising CB2 molecules have been designed, synthesized and preliminary evaluated. The next step would be to study them (bioactivity) and their derivatives further. Not much speculation is needed at this point.
The study is technically very well performed and the methods etc. have been described in sufficient details. The manuscript is very well written in excellent language. The results are predominantly presented very clearly (an exception is Table 1, see Major points). The results are very clear for drawing the conclusions.
Major points:
- Table 1 does not represent (directly) binding affinities of the compounds, so it would be more appropriative to use a title “Displacement of [3H]CP-55,940 from CB1 and CB2 receptors by 10 µM concentration of compounds 4a-4q.” or a similar one that exactly expresses what has been done.
Please explain clearly what the values in Table 1 mean (the amount of specific [3H]CP-55,940 binding displaced (in %)) also in the text of Table 1.
The values of 4d and 4m in Table 1 are now in two rows. Please correct this.
- In page 11, Radioligand binding or displacement of radioligand binding assay is usually not (and not the one used in this manuscript) an in vitro receptor activity assay. It does not measure receptor activity and it does not even tell whether the displacing compound is a full or partial agonist, antagonist or inverse agonist. It only shows that the displacing compound binds to the same site as the radioligand and shows the affinity, Ki (in case of competitive displacement) or allosterically displaces the radioligand (in case of allosteric displacement). Please use the expression “in vitro CB1 and CB2 binding assays” or “displacement of [3H]CP-55,940 binding assay” instead of “in vitro CB1 and CB2 activity assays”.
- The last sentence in page 11 before Table 1 states that “no significant binding was observed for CB1 (Table2)”. Compounds 4k and 4q displace 41.26% and 55.27% of the binding at 10 µM concentration, respectively. Is it really so, that they did not displace any binding in Table 2 experiments? What were the concentration ranges of 4k and 4q used in Table 2 experiments.
Minor points:
- It would be good to add a conclusive sentence or two in abstract. Now it ends in a summary of the main results.
- In 1. Experimental Methods, compound 4n: Na2SO4: numbers should be subscripts
- Page 8, row 4: reconstitutded -> reconstituted
- Page 11, bottom: Subtitle “In vivo PK studies of the lead analog 4k” should be in italics.
- Page 14, Conclusions: There is a jump from row 5 to row 6 in the middle.
Author Response
Reviewer 3:
Comment 1: The study is technically very well performed and the methods etc. have been described in sufficient details. The manuscript is very well written in excellent language. The results are predominantly presented very clearly (an exception is Table 1, see Major points). The results are very clear for drawing the conclusions.
Response 1: We appreciate the reviewer’s positive comment.
Comment 2: Table 1 does not represent (directly) binding affinities of the compounds, so it would be more appropriative to use a title “Displacement of [3H]CP-55,940 from CB1 and CB2 receptors by 10 µM concentration of compounds 4a-4q.” or a similar one that exactly expresses what has been done. Please explain clearly what the values in Table 1 mean (the amount of specific [3H]CP-55,940 binding displaced (in %)) also in the text of Table 1.
Response 2: The title of table 1 has been changed to “Displacement of [3H]CP-55,940 from CB1 and CB2 receptors by 10 µM concentration of compounds 4a-4q” as suggested. The values in table 1 have been clarified in the text.
Comment 3: The values of 4d and 4m in Table 1 are now in two rows. Please correct this.
Response 3: Table 1 has been adjusted as suggested.
Comment 4: In page 11, Radioligand binding or displacement of radioligand binding assay is usually not (and not the one used in this manuscript) an in vitro receptor activity assay. It does not measure receptor activity and it does not even tell whether the displacing compound is a full or partial agonist, antagonist or inverse agonist. It only shows that the displacing compound binds to the same site as the radioligand and shows the affinity, Ki (in case of competitive displacement) or allosterically displaces the radioligand (in case of allosteric displacement). Please use the expression “in vitro CB1 and CB2 binding assays” or “displacement of [3H]CP-55,940 binding assay” instead of “in vitro CB1 and CB2 activity assays”.
Response 5: The expression “in vitro CB1 and CB2 binding assays” has been used instead of “in vitro CB1 and CB2 activity assays” as suggested.
Comment 6: The last sentence in page 11 before Table 1 states that “no significant binding was observed for CB1 (Table2)”. Compounds 4 and 4q displace 41.26% and 55.27% of the binding at 10 µM concentration, respectively. Is it really so, that they did not displace any binding in Table 2 experiments?
Response 6: The last sentence in page 11 before table 1 has been adjusted. Compounds 4k and 4q have not exhibited strong displacement towards CB1 (>60%) in the initial binding assays to acquire their Ki towards CB1.
Comment 7: It would be good to add a conclusive sentence or two in abstract. Now it ends in a summary of the main results.
Response 7: A conclusive statement has been added to the abstract as suggested.
Comment 8: In 1. Experimental Methods, compound 4n: Na2SO4: numbers should be subscripts.
Response 8: The numbers have been adjusted as suggested.
Comment 9: Page 8, row 4: reconstitutded -> reconstituted.
Response 9: Corrected as suggested.
Comment 10: Page 11, bottom: Subtitle “In vivo PK studies of the lead analog 4k” should be in italics.
Response 10: Page 11, bottom: Subtitle “In vivo PK studies of the lead analog 4k” has been italicized as suggested.
Comment 11: Page 14, Conclusions: There is a jump from row 5 to row 6 in the middle.
Response 11: The extra space in the conclusion has been removed as suggested.
Round 2
Reviewer 2 Report
Please delete the descriptions related with tissues’ Vd in the main text.
Author Response
Comment 1: Please delete the descriptions related with tissues’ Vd in the main text.
Response 1: The descriptions related with tissues’ Vd in the main text has been deleted as suggested.